# Secure Outlier-Aware Large Language Model Inference

**Lifan Zhao**
Shanghai Qi Zhi Institute
zhaolf@sqz.ac.cn

**Zhixuan Fang** *
Tsinghua University
Shanghai Qi Zhi Institute
zfang@mail.tsinghua.edu.cn

## ABSTRACT

Secure multiparty computation allows the client to secretly inference their sensitive inputs without acquiring the proprietary machine learning model weights. As the decoder-only transformer-based large language model becomes the popular paradigm, the desire of applying MPC in large language models is increasing. However, such inference usually leads to great amount of latency, which is due to nonlinear operations in the Transformer architecture. Recent works either focus on improving cryptographic primitives or re-architecting and re-training to make LLM MPC-friendly. We, on the other hand, observe that properly addressing outlier phenomena, which are unique yet universal properties existing across different LLMs, can effectively reduce the input domain and thereby design faster protocols for non-linear operations. Hence, we propose **S**ecure **O**utlier-**A**ware **L**arge Language Model Inference framework (SOAL), which accelerates the RMSNorm operation by nearly $2\times$, SiLU by $2\times$, and Softmax by more than $5\times$. SOAL maintains similar performance as the original model without any fine-tuning requirement.

## 1 INTRODUCTION

As large language models (Radford et al., 2019; Gupta et al., 2022; Jiang et al., 2023) have demonstrated remarkable capability in variety of aspects, the concern of leakage of private sensitive information to cloud-based LLM inference services is increasingly raised. On one hand, precious business data, and sensitive personal data are processed by model providers' servers without any privacy protection. On the other hand, LLM model weights, which are the product of trillions of tokens and millions of GPU-hours of training, are uploaded and stored on third-party servers, counting on their pledge. Secure multiparty computation (MPC) offers a solution where LLM inference services can be offered without leaking any information to either users or model providers. However, making the LLM inference with an MPC protocol is notably slow. Inferencing a 64-token input once using Llama2-7B (Gupta et al., 2022) via CrypTen (Knott et al., 2021) would take 169.76 seconds, while it only takes less than one second in plain text. Such latency becomes even greater as the input sequences gets longer, i.e., it would take more than 428.95 seconds for a 512-token input. Such inefficiency stems from preprocessing-based MPC frameworks' difficulty in handling nonlinear layers, including Softmax, Activation (SiLU or GeLU), and Normalization (Layernorm or RMSNorm).

The inherent complexity of non-linear protocols comes from the disproportionate output growth across a large input domain. Meanwhile, activations during LLM inference may vary by many orders of magnitude. For example, activations for normalization layers in Llama2-7B (Gupta et al., 2022) vary from $10^{-5}$ to $10^3$, as shown in Figure 5b. For non-linear protocols covering such a wide input domain, directly using a lookup table (LUT) protocol would require a LUT with 32-bitwidth. As LUT protocols usually have a complexity of $O(2^n)$, it is very expensive to compute. Other methods, like Goldschmidt's method (Goldschmidt, 1964) would similarly consequent in more iterations. Observing such flaws, previous works (Mohassel & Zhang, 2017; Cheng et al., 2023; Luo et al., 2024) have tried to substitute non-linear operations with MPC-friendly ones. SecureML (Mohassel

---

*Corresponding author.

& Zhang, 2017) used Softmax$(x) = \frac{\text{ReLU}(x)}{\sum \text{ReLU}(x_i)}$ to avoid exponential computation. MPCFormer (Cheng et al., 2023) substituted GeLU$(x)$ with Quad $= 0.125x^2 + 0.25x + 0.5$. Although they exploited the robustness of LLMs such that the model may still be functional after substitution, these methods face substantial accuracy degradation and require additional fine-tuning to compensate for the quality discrepancy. Such methods modify the original model, induce training overhead, and bring doubt regarding in the validity and quality of the new model.

Motivated by above challenges, we ask the following research question:

*Can we achieve efficient non-linear protocols specifically designed for MPC LLM inference?*

Meanwhile, another line of recent research offers us a critical insight: that LLMs, especially decoder-only transformer-based LLMs, have many unique properties shared in common, which have been utilized in a variety of domains. Dettmers et al. (Dettmers et al., 2022) observed that most activations and model weights have a skewed distribution, which means their values are concentrated in a narrow range and only very few activations are significantly large. By handling outliers separately, they succeeded in making inference with FP8, a one-byte data format for floating point number, rather than FP32. This method effectively lowers the requirement for GPU memory without sacrificing much performance. Inspired by their observation, we study non-linear layers and find that similar outlier distribution phenomenon also persists. Figure 1 demonstrates the layer-wise activation distribution for normalization layers, where most activations aggregate around a certain value with small variances. Meanwhile, there are a few outlier activations that deviate far from the majority. It is the existence of such outliers that increases the complexity of MPC non-linear protocols.

**Our approach and contributions.** We propose **S**ecure **O**utlier-**A**ware **L**arge Language Model Inference framework (SOAL), a two-stage framework which addresses the unique outlier features in the decoder-only transformer-based LLMs and optimizes non-linear protocols without compromising LLM performance. In detail, our contribution is threefold:

1. We discover different outlier phenomena existing in activations for non-linear layers across various LLMs. We further address such phenomena by prefixing special tokens to users' input prompt to manipulate the occurrence of outliers. Our evaluation shows that such modification brings no performance degradation.

2. We design new protocols for normalization and activation layers, which benefit from the control of outliers. Based on a collateral feature from prefixing tokens, we improve the Softmax protocol as well. Our protocols can work with different cryptographic schemes.

3. Our SOAL achieves nearly $2\times$ speedup when inferencing Llama2-7B and GPT-2 with 512 tokens under the additive secret sharing scheme, while maintaining the accuracy of the output. SOAL also obtain efficiency improvement on both online time and total keysize under the function secret sharing scheme.

## 2 RELATED WORK

### 2.1 PRIVACY-PRESERVING MACHINE LEARNING.

MPC originates from the Billionaire problem (Yao, 1986; Shamir, 1979) whose goal is to enable multiple untrusted parties to jointly compute a function while keeping the inputs private. As LLMs show their incredible ability, recent works focus on mitigating MPC with LLM inference only. They primarily follow two complementary directions. The first focuses on accelerating secure inference through advanced cryptographic primitives. For instance, systems such as SIGMA (Gupta et al., 2024) and BOLT (Chen et al., 2024b) leverage function secret sharing (FSS) and vector oblivious linear evaluation (VOLE), respectively, to reduce communication and latency. Others, like Iron (Hao et al., 2022), PrivFormer (Dalskov et al., 2023), and PUMA (Dong et al., 2025), optimize secure matrix multiplications and nonlinear layers using a mix of secret sharing, homomorphic encryption, and polynomial approximations, demonstrating practical performance on full-scale transformer models. Innovations like BumbleBee (Huang et al., 2025), SHAFT (Kei & Chow, 2025), and NEXUS (Zhang et al., 2025) further improve scalability by introducing protocol-level optimizations for core transformer components, including attention, GELU, and Softmax. In parallel, a second thread of research modifies transformer architectures to better suit MPC. Works such as

MPCFormer (Cheng et al., 2023) and SecFormer (Luo et al., 2024), based on additive secret sharing (ASS) scheme, substitute costly nonlinearities (e.g., Softmax and GELU) with MPC-friendly alternatives like low-degree polynomials and train these altered models using knowledge distillation, achieving significant efficiency gains with minimal accuracy loss. However, the requirement of knowledge distillation may change the model weights, thereby hindering its broader usage.

### 2.2 OUTLIERS IN LLM ACTIVATION

In this work, we specifically focus on decoder-only transformer-based large language models (Radford et al., 2019; Gupta et al., 2022; Jiang et al., 2023). During LLM inference, the activations for each layer, no matter what input prompt is, always contain a few activations, known as outliers, that stray far from others. On the other side of the coin, the rest of the activations exhibit a skewed pattern concentrated in a narrow range. Outliers in LLM inference have drawn much attention recently (Bondarenko et al., 2021; Wei et al., 2022; Puccetti et al., 2022; Dettmers et al., 2022; Xiao et al., 2023b; Sun et al., 2024). Some of these activations are related to particular features, which are known as outlier features. Puccetti et al. (Puccetti et al., 2022) showed empirically that the occurrence of outlier features is related to the frequency of tokens in the training distribution. Dettmers et al. (Dettmers et al., 2022) showed that the scale of an LLM is related to the emergent properties of these outlier features, and utilized such features to effectively quantize model weights. Other activations may jointly pertain to specific channels and tokens. Sun et al. (Sun et al., 2024) introduced the concept of "massive activations" — rare but extremely large activation values that appear consistently in special tokens (e.g., at begin of sentence (*bos*) or newline tokens). Chen et al. (Chen et al., 2024a) further utilize such a relationship to propose a prefixing method and make INT4 quantization comparable to other dynamic per-token quantization methods. Besides linear layers' activations, (Xiao et al., 2023a;b; Bondarenko et al., 2023) reveal the attention sink phenomenon, where those related to the *bos* token tend to be larger. In this work, we extend the study to non-linear layers and reveal that such outliers not only persist, but also can be used to optimize MPC protocols.

## 3 OUTLIERS IN NON-LINEAR LAYERS

In this section, we extend the LLM outlier study from the linear layer to non-linear operations in transformer-based LLMs, SoftMax, Normalization and Activation. Given an input activation for non-linear operations, outliers are a small group of values which greatly deviate from the average. Outliers require a larger input domain for non-linear MPC protocols, which consequently need either more bits to handle or more iterations to approximate.

We use RedPajama (Weber et al., 2024), a dataset released after the chosen LLMs, to run 128 inferences with different token lengths and collect activations for different non-linear operations. We conduct additional experiments using more LLMs and different datasets in Appendix A. We identify different outlier phenomena corresponding to different non-linear layers. Such outlier phenomena exist across various datasets, showing that these are ubiquitous features belonging to the decoder-only transformer-based LLMs.

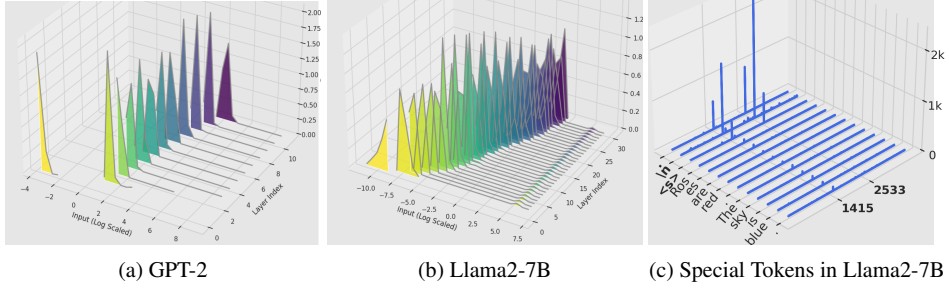

|   (a) GPT-2   |   (b) Llama2-7B   |   (c) Special Tokens in Llama2-7B   |

Figure 1: The activations distribution for normalization operation in (a) GPT-2 and (b) Llama2-7B. We use bold line for better visualizing the deviation from majority values. (c) Special tokens are closely related to the outliers.

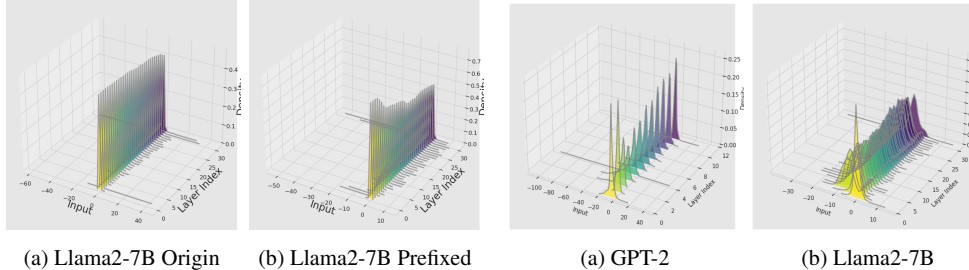

| (a) Llama2-7B Origin | (b) Llama2-7B Prefixed | (a) GPT-2 | (b) Llama2-7B |

Figure 2: Activation Distribution for SiLU.      Figure 3: Activation Distribution for SoftMax.

**Normalization** We visualize the distribution of activations for each normalization layer in Figure 1. The long tails in most layers indicate that there exists a small group of activations whose values greatly depart from the majority. It is the existence of these outliers which greatly extend the domain requirement for reciprocal square root protocol design. Fortunately, the occurrence of these outliers shows a strong connection with special tokens. As shown in Figure 1c, in Llama2-7B, outliers locates at the positions where the token "." (period), "\n" (newline) and "⟨s⟩" (begin of sentence (BOS)) appear. Such observations align with the outlier findings in linear layers from previous works (Sun et al., 2024; Xiao et al., 2023a). We will describe the detail method to extract special tokens and address these *Special-Token Outlier* later in Section 4.1.1.

**Activation** We display the input distribution for Activation layer in Figure 2a. Values aggregate and form spikes around zero where the activation function is most sensitive, while outliers appear in the first and last few layers. We additionally show that outliers in activation layers are also affected by the aforementioned special tokens in Figure 2b.

**SoftMax** We further explore the layerwise activations distribution for SoftMax in Figure 3. The majority of activation values concentrate around their layerwise average values, which also matches the observation in FlashDecoding++(Hong et al., 2024) that $> 99.99\%$ $x_i$ are within a certain range. We will later demonstrate that the outliers can be effectively captured by *Conformant Maxima*—the local maxima we discovered, which are derived from values gathered from predefined locations within attention logits.

In conclusion, these different yet consistent observations on non-linear layers point toward an answer to the question posed earlier, that if we can address these outliers properly, the skewed distribution of the rest activations can be utilized to design better protocols for these non-linear operations.

## 4 METHOD

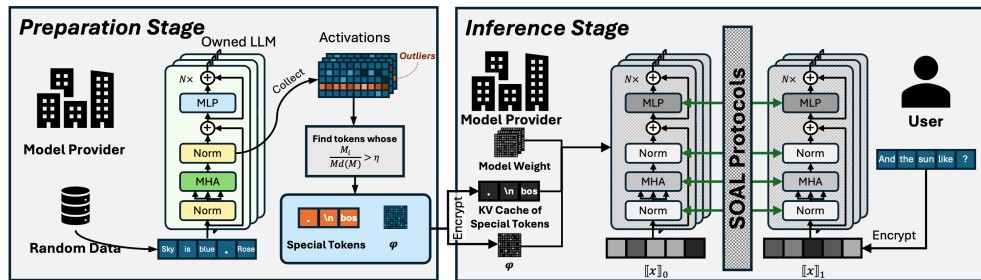

Figure 4: SOAL Framework. In preparation stage, model providers collect activations by performing extra inferences with their LLMs. They then obtain special tokens and auxiliary parameters $\varphi$ from these activations. In the inference stage, model provider and user follow our new protocols to efficiently perform MPC LLM inference.

SOAL focuses on the standard two-party computation (2PC) with one trusted dealer (Beaver, 1991; Bendlin et al., 2011; Damgård & Zakarias, 2013) that has received significant attention in privacy-

preserving LLM inference (Gupta et al., 2022; Knott et al., 2021; Jawalkar et al., 2024; Luo et al., 2024; Kei & Chow, 2025). That is, there are two semi-honest (honest but curious) parties, $P_0$ as the user who has the input prompt $x_0$ and $P_1$ as the model provider who offers the inference service with the model weights $x_1$. In the offline phase, a trusted dealer offers correlated randomness to accelerate online inference. Together, the 2PC want to perform the computation of a public function $y = F(x_0, x_1)$ without revealing anything more than the function output $y$ to each other. Detailed preliminaries about MPC and LLM are provided in Appendix B.

In general, SOAL is a two-stage framework as shown in Figure 4. In the preparation stage, as detailed in Section 4.1, the model provider utilizes previous observations on outliers in non-linear layers and prepares model-specific auxiliary parameters used for secure inference. In the inference stage, which we describe fully in Section 4.2, the model provider and the user follow the new designed non-linear protocols to perform faster secure LLM inferences.

## 4.1 PREPARATION STAGE: ADDRESS OUTLIER

### 4.1.1 SPECIAL-TOKEN OUTLIER

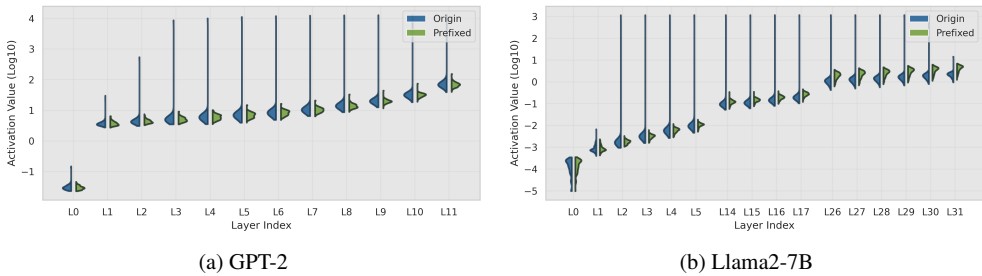

(a) GPT-2            (b) Llama2-7B

Figure 5: Comparison of layer-wise normalization activation distribution in violin plot. Blue half represents for original distribution, and green part is the distribution after prefixing special tokens.

As mentioned above, the outliers in normalization layers not only exist, but also occur along with special tokens. To identify the special tokens, we use the activations for pre-layer normalization and acquire the special-token outliers as follow:

We collect activations $X \in \mathbb{R}^{N \times L \times F}$ through $N$ inference for each layer. We then statistic token-wise maximum value, $M \in \mathbb{R}^T$, where $T$ is the tokenizer size. For $i$-th token whose ratio of its maximum value over the median maximum value of all other tokens exceeds a threshold $\eta$, such as $\frac{M_i}{\text{median}(M)} > \eta$, we record the corresponding token as a special token. We set $\eta = 8$ here since we want to shrink the input domain as much as possible.

Now, with the special tokens at hand, we can address these outliers in normalization layers. Since the outliers appear where these special tokens first emerge, we prefix these tokens ahead of user's input prompts to manipulate such outliers appearing only at the prefixed positions. To visualize the effect, we inference twice with the same prompt, except that in the second time, we prefix these special tokens ahead of that prompt. As shown in Figure 5, after prefixing the special tokens, the green parts no longer have long-tails in any layers, which means that the outliers have been limited within those prefixed positions. What's more, due to LLM's autoregressive property, these special tokens can be stored in the format of KV-cache (Radford et al., 2019) to avoid redundant computation, and use as the model-specific auxiliary parameters during inference stage. Compared to the overhead of fine-tuning the entire model for different tasks, such a token-identifying process can be easily done by the model provider offline within a few minutes.

### 4.1.2 CONFORMANT MAXIMA

Outliers in SoftMax conform with a pattern that we can utilize. We observe that the maxima of softmax input present a connection with special tokens. Given the collected softmax inputs $x \in \mathbb{R}^{N \times L \times L}$ from $N$ inferences of $L$-token length prompts, we present a heatmap with the shape $L \times L$ in Figure 6. Each grid therefore represents for one position in $x$. We then use the grayscale to

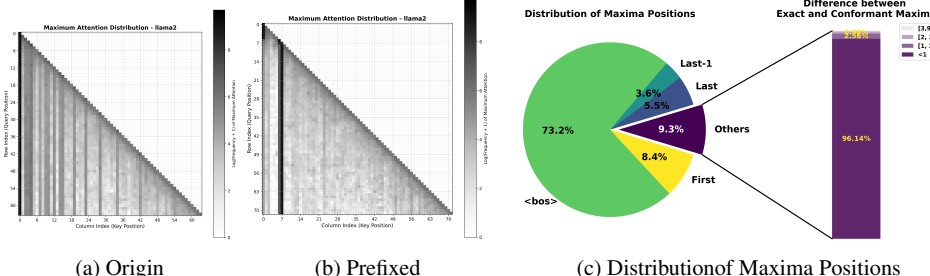

(a) Origin        (b) Prefixed        (c) Distributionof Maxima Positions

Figure 6: (a) Heatmap of origin maxima positions of Llama2-7B. Grayscale means the times maxima appear at that position. (b) Heatmap with prefixed special tokens. (c) Pie chart of the occurrence of exact maxima. For rest of maxima locating in other positions, conformant maxima are close to them.

---

**Algorithm 1:** Secret Reciprocal Square Root, $\Pi_{\text{rsqrt}}$

**Input:** For $\sigma \in \{0,1\}$, $P_\sigma$ holds the shares $[\![x]\!]_\sigma$

**Output:** For $\sigma \in \{0,1\}$, $P_\sigma$ holds the shares $[\![y]\!]_\sigma$ where $y = \text{RMSNorm}(\mathbf{x})$.

1   $[\![x']\!]_\sigma := [\![x]\!]_\sigma - 1$
2   $[\![x_2']\!]_\sigma := \Pi_{\text{square}}([\![x']\!]_\sigma)$
3   $[\![x_3']\!]_\sigma := \Pi_{\text{mul}}([\![x']\!]_\sigma, [\![x_2']\!]_\sigma)$
4   $[\![y]\!]_\sigma := a + b[\![x']\!]_\sigma + c[\![x_2']\!]_\sigma + d[\![x_3']\!]_\sigma$
5   **foreach** *iter < 2* **do**
6     $\lfloor \; [\![y]\!]_\sigma := [\![y]\!]_\sigma \cdot (3 - [\![x']\!]_\sigma \cdot [\![y']\!]_\sigma^2)/2$
7   **return** $[\![y]\!]_\sigma$;

---

**Algorithm 2:** Secret RMSNorm, $\Pi_{\text{RMSNorm}}$

**Input:** For $\sigma \in \{0,1\}$, $P_\sigma$ holds the input share $[\![x]\!]_\sigma$, and scale share $[\![\varphi_i]\!]_\sigma$

**Output:** For $\sigma \in \{0,1\}$, $P_\sigma$ holds the shares $[\![y]\!]_\sigma$ where $y = \text{RMSNorm}(\mathbf{x})$.

1   $[\![x']\!]_\sigma := \Pi_{\text{mul}}([\![x]\!]_\sigma, [\![\varphi_i]\!]_\sigma)$
2   $[\![x']\!]_\sigma := \Pi_{\text{square}}([\![x']\!]_\sigma)$
3   $[\![d]\!]_\sigma := \sum([\![x']\!]_\sigma) + \epsilon$
4   $[\![d']\!]_\sigma := \Pi_{\text{rsqt}}([\![d]\!]_\sigma)$
5   $[\![y]\!]_\sigma := \Pi_{\text{mul}}([\![d']\!]_\sigma, [\![x']\!]_\sigma)$.
6   **return** $[\![y]\!]_\sigma$;

---

represent the number of times maxima locate at that position. Without prefixing, maxima may appear where special tokens first locate; therefore, there are some random vertical lines in Figure 6(a). Meanwhile, with special tokens prefixed, the maxima concentrate at the position of "$\langle\text{bos}\rangle$" tokens and the first one and last two tokens of each row. Such phenomenon has been identified as the attention sink, by StreamLLM (Xiao et al., 2023b). They pointed out that certain token may receive more attention than others due to the occurrence of these tokens in the training datasets. Such phenomenon gives us a potential way to simplify maximum finding operation in SoftMax.

We record the exact maximum values $x_{\max} \in \mathbb{R}^{N \times H \times L}$ of softmax inputs from $N$ different inferences. We then collect the activations at (1) where the *bos* token locates, (2) the first token position, and (3) the last two positions of each row. The local maxima are selected from these activations, and we call them *Conformant Maxima*. As shown in Figure 6c, over 90% exact maxima's positions fall in these predefined positions. What's more, for those maxima mismatched, the differences between the exact maxima and our conformant maxima are also small. Hence, by prefixing special tokens and collecting conformant maxima, we can avoid the $O(n \log n)$ computation to find the exact maximum over an $n$-token length input.

### 4.2 INFERENCE STAGE: OPTIMIZE MPC PROTOCOLS

The bond between addressing outliers and optimizing MPC protocols lies in input domain reduction. In this subsection, we demonstrate our new non-linear protocols under ASS scheme. The protocols under FSS scheme will be introduced in Appendix E.

#### 4.2.1 NORMALIZATION

Normalization is used between multi-head attention layers and feed-forward layers to keep activations from exploding. For the input $X \in \mathbb{R}^{L \times F}$, where $F$ is the feature dimension, RMSNorm

**Algorithm 3:** Secret Sigmoid for LLM, $\Pi_{\text{sigmoid}}$

**Input:** For $\sigma \in \{0,1\}$, $P_\sigma$ holds the input share $[\![x]\!]_\sigma$, and $[\![\nu_0]\!]_\sigma, [\![\nu_1]\!]_\sigma$.
**Output:** For $\sigma \in \{0,1\}$, $P_\sigma$ holds the shares $[\![y]\!]_\sigma$ where $y = \sigma(x)$.

1   $[\![x]\!]_\sigma := -\log_2 e \cdot [\![x]\!]_\sigma$
2   $[\![x_i]\!]_\sigma := \Pi_{\text{trunc}}(x')$, $[\![x_f]\!]_\sigma := [\![x]\!]_\sigma - [\![x_i]\!]_\sigma$
3   $[\![\nu_0]\!]_0 = [\![\nu_0]\!]_0 + 2^{x_{f_0}}$, $[\![\nu_1]\!]_1 = [\![\nu_1]\!]_1 + 2^{x_{f_1}}$
4   $[\![v_2]\!]_\sigma := \Pi_{\text{mul}}[\![\nu_0]\!]_\sigma, [\![\nu_1]\!]_\sigma$
5   $[\![v_1]\!]_\sigma, [\![\overline{v_1}]\!]_\sigma = \Pi_{\text{LUT}}([\![x_i]\!]_\sigma)$
6   $[\![v_2]\!]_\sigma := \Pi_{\text{mul}}([\![v_2]\!]_\sigma, [\![\overline{v_1}]\!]_\sigma)$
7   $[\![\delta]\!]_\sigma := v_1 + v_2$
8   $[\![\alpha]\!]_\sigma := [\![e - f\delta + g\delta^2]\!]_\sigma$
9   $[\![r]\!]_\sigma := 2 \cdot [\![\alpha]\!]_\sigma - \Pi_{\text{mul}}([\![\delta]\!]_\sigma, \Pi_{\text{square}}([\![\alpha]\!]_\sigma))$
10   $[\![y]\!]_\sigma := \Pi_{\text{mul}}([\![r]\!]_\sigma, [\![v_1]\!]_\sigma)$
11   **return** $[\![y]\!]_\sigma$;

**Algorithm 4:** Secret Softmax in LLM, $\Pi_{\text{softmax}}$

**Input:** For $\sigma \in \{0,1\}$, $P_\sigma$ holds the shares $[\![x]\!]_\sigma$.
**Output:** For $\sigma \in \{0,1\}$, $P_\sigma$ holds the shares $[\![y]\!]_\sigma$ where $y = \text{softmax}(x)$.

1   $[\![\tau]\!]_\sigma = \Pi_{max}([\![x_{[b]}]\!]_\sigma, [\![x_{[i-1]}]\!]_\sigma, [\![x_{[i]}]\!]_\sigma, [\![x_{[0]}]\!]_\sigma)$
2   $[\![x']\!]_\sigma := [\![x]\!]_\sigma - [\![\tau]\!]_\sigma$
3   $[\![e]\!]_\sigma := \Pi_{\exp}([\![x']\!]_\sigma)$
4   $[\![d]\!]_\sigma := \sum [\![e]\!]_\sigma$
5   $[\![m]\!]_\sigma, [\![\bar{m}]\!]_\sigma := \Pi_{\text{msb}}(d)$
6   $[\![d']\!]_\sigma = \Pi_{\text{mul}}([\![d]\!]_\sigma, [\![m]\!]_\sigma)$
7   $[\![d']\!]_\sigma := \Pi_{\text{NR}}([\![d']\!]_\sigma)$, where the initial guess is $1.45 - 0.5[\![d']\!]_\sigma$. // Two NR iterations.
8   $[\![y]\!]_\sigma := \Pi_{\text{mul}}([\![e]\!]_\sigma, \Pi_{\text{mul}}([\![d']\!]_\sigma, [\![\bar{m}]\!]_\sigma))$.
9   **return** $[\![y]\!]_\sigma$;

of each row $\mathbf{x}$ is calculated with $\gamma \cdot \frac{\mathbf{x}}{\sqrt{\frac{1}{F}\sum_{j=1}^{F}(x_j)^2+\epsilon}} + \beta$, where $\gamma, \beta$ are fixed parameters. The complexity lies in the reciprocal square root function which need multiple iterations to approximate.

We demonstrate our new protocol based on an approximation algorithm in Algorithm 2. As shown in Figure 5, the activations concentrate closely around their layerwise centers after the special-token outliers have been constrained in the prefixed token of KV-cache. Such concentration centers shift according to specific layers, from a scale of $10^{-5}$ to $10^{1}$. To further shrink the input domain, we scale the activations with layerwise secret values $[\![\varphi]\!]_\sigma$, and such scaling can later be canceled out via the division. Thus, we change the RMSNorm($\mathbf{x}$) to $\gamma \cdot \frac{\varphi_i \cdot \mathbf{x}}{\sqrt{\frac{1}{F}\sum_{j=1}^{F}(\varphi_i \cdot x_j)^2+\epsilon}} + \beta$. Now as the input domain for the reciprocal square root has been reduced, we can use a third-degree polynomial centering at $x = 1$ to get the initial guess. The coefficients $a, b, c, d$ are acquired by BFGS (Broyden, 1970) with MSE loss, and are set to $a = 0.913389, b = -0.860195, c = 1.028723, d = -0.359165$ in this work. With the proper initial guess, we can use only two Newton-Raphson iterations to obtain the precise result.

### 4.2.2 ACTIVATION

Activation is designed to bring non-linearity to LLM models. For the input $X \in \mathbb{R}^{L \times F}$, the SiLU function is calculated with $\text{SiLU}(x) = x \cdot \sigma(x)$, where $\sigma(x) = \frac{1}{1+e^{-x}}$ is the sigmoid function.

As mentioned above, the outliers of SiLU inputs exhibit a connection to the special tokens. As shown in Figure 2b, with prefixed special tokens, the range of input domain becomes acceptable for efficient MPC protocol design. Again, the calculation of prefixed outliers can be omitted by the KV-cache. We now reformulate the sigmoid as $\sigma(x) = \frac{2^{-x_i \cdot (1-\xi)}}{2^{-x_i \cdot (1-\xi)} + 2^{x_f} \cdot 2^{x_i \cdot \xi}}$, where $x_i, x_f$ represent the integer and fraction part of $x$, and the sign $\xi = \mathbf{1}\{x < 0\}$ is 1 if $x$ is less than zero, otherwise 0. Hence, we implement the sigmoid protocol in Algorithm 3.

We first convert the exponential base to 2 and use local truncation protocol to get fraction $[\![x_f]\!]_\sigma$ and integer parts $[\![x_i]\!]_\sigma$. As the fraction part is small, each party can use the owned share $x_{f_\sigma}$ to calculate a local exponential result $2^{x_{f_\sigma}}$ which can be further re-shared secretly with the two predefined zero secret share $[\![\nu_0]\!]_\sigma$ and $[\![\nu_1]\!]_\sigma$. With one multiplication, we obtain the $[\![2^{x_f}]\!]_\sigma$. Now, we can further simply the equation as $\frac{v_1}{v_1 + v_2 \cdot \overline{v_1}}$, where $v_1 = 2^{-x_i \cdot (1-\xi)}$, $\overline{v_1} = 2^{x_i \cdot \xi}$ and $v_2 = 2^{x_f}$. As $[\![x_i]\!]_\sigma$ is an integer with small input domain, we can use LUT protocol to get $v_1$ and $\overline{v_1}$, both of which depend on the integer part. Then, in Lines 6-7, we obtain the denominator by adding $v_1$ and $v_2 \cdot \overline{v_1}$ together, whose range lies between $(1, 2.5)$. Given the denominator's domain is small, we use a second-degree polynomial approximation to get a decent initial guess of the reciprocal in Line 7, where $e = 1.830181, f = 1.078845, g = 0.205469$. At last, we need one more round of Newton-Raphson iteration and multiply $v_1$ back to get the sigmoid result.

### 4.2.3 SOFTMAX

Softmax is the key element in Transformer structure. Given $X \in \mathbb{R}^{L \times L}$ where $L$ is the token length, the softmax of $\mathbf{x} = \{x_0, x_1, ..., x_i\}$, the $i$-th row of $X$, is calculated with $\text{softmax}(\mathbf{x}) =$

---

**Algorithm 5:** Secret Exponetial, $\Pi_{\exp}$

---

**Input:** For $\sigma \in \{0, 1\}$, $P_\sigma$ holds the shares
  $[\![x]\!]_\sigma$, and $[\![\nu_0]\!]_\sigma$, $[\![\nu_1]\!]_\sigma$.
**Output:** For $\sigma \in \{0, 1\}$, $P_\sigma$ holds the shares
  $[\![y]\!]_\sigma$ where $y = \exp(x)$.

1 $[\![x']\!]_\sigma := \log_2 e \cdot [\![x']\!]_\sigma$
2 $[\![x_i]\!]_\sigma := \Pi_{\text{trunc}}([\![x']\!]_\sigma, s)$,
  $[\![x_f]\!]_\sigma := [\![x']\!]_\sigma - [\![x_i]\!]_\sigma$
3 $[\![\nu_0]\!]_0 := [\![\nu_0]\!]_0 + 2^{x_{f_0}}$, $[\![\nu_1]\!]_1 := [\![\nu_1]\!]_1 + 2^{x_{f_1}}$
4 $[\![m]\!]_\sigma := \Pi_{\text{Mul}}([\![\nu_0]\!]_\sigma, [\![\nu_1]\!]_\sigma)$
5 $[\![e]\!]_\sigma := \Pi_{\text{LUT}}([\![x_i]\!]_\sigma)$
6 $[\![y]\!]_\sigma := \Pi_{\text{Mul}}([\![m]\!]_\sigma, [\![e]\!]_\sigma)$
7 **return** $[\![y]\!]_\sigma$;

---

Figure 7: GPT-2 Inference Speed Comparison

| Model | Method | SoftMax | | Normalization | | Activation | | Total | |
|---|---|---|---|---|---|---|---|---|---|
| | | Time (s) | Comm.(GB) | Time (s) | Comm.(GB) | Time (s) | Comm.(GB) | Time (s) | Comm.(GB) |
| GPT-2 | CrypTen | 30.79 | 52.54 | 3.07 | **0.88** | 10.89 | 19.68 | 48.86 | 76.79 |
| | SOAL | **6.80** | **4.52** | **2.99** | 1.18 | **9.17** | **15.26** | **23.30** | **24.64** |
| Llama2-7B | CrypTen | 199.99 | 375.03 | 27.86 | **12.25** | 87.30 | 169.31 | 428.95 | 702.44 |
| | SOAL | **26.62** | **32.13** | **14.87** | 16.01 | **37.13** | **67.19** | **193.59** | **261.15** |
| Mixtral 8x7B | CrypTen | 242.98 | 376.01 | 65.86 | **24.31** | 264.52 | 441.00 | 1104.46 | 1611.20 |
| | SOAL | **39.66** | **32.45** | **31.25** | 32.25 | **104.44** | **161.11** | **668.23** | **984.50** |

Table 1: Time and communication cost for secure inference of 512-token prompt. Average of 5 runs.

$\frac{\exp(\mathbf{x} - \max(\mathbf{x}))}{\sum_{j=1}^{i} \exp(x_j - \max(\mathbf{x}))}$. Among the three obstacles in calculating Softmax under MPC setting — maximum, exponential and reciprocal — finding the maximum is the most unique one, as its communication cost grows $O(n \log n)$ log-linearly according to the actual input length $n$.

The common practice of subtracting the exact maxima is to avoid numerical exploding problem caused by backpropagation during training, but, the maxima can mathematically be replaced by any arbitrary number $\tau$ during inference. As mentioned in Section 4.1.2, conformant maxima matches exact maxima for most of the time, meanwhile the difference is small when mismatched. It is proper candidate for $\tau$ to avoid the log-linear complexity of exact maximum finding. Hence, our softmax protocol can now handle any length of input, always with fixed communication cost and rounds.

We present our protocol for Softmax in Algorithm 4. Noticing that the number of reciprocal operation is much smaller than that of exponential, we directly get the most significant bits of the denominators. Therefore, by multiplying $m$ with denominator $d$, we scale the denominator to a small domain where the reciprocal can be obtained with two NR iterations.

As our conformant maxima no longer promises the non-positive property of the SoftMax input, we display a new exponential protocol for the secret exponential function that can handle both positive and negative values in Algorithm 5. We first local truncate the $[\![x]\!]_\sigma$ by precision bits plus extra $s$ bits to get $[\![x_i]\!]_\sigma$ and $[\![x_f]\!]_\sigma$, where $x = x_i \cdot 2^{f+s} + x_f$. Due to the product of powers rule, each party can locally calculate the exponential result of their share $x_f$ at hand. Afterward, they can secretly reshare the result $2^{x_{f_\sigma}}$ to $[\![\nu_0]\!]_\sigma$ or $[\![\nu_1]\!]_\sigma$. With one more multiplication, we can get correct result of $2^{x_f}$. Then, with the help of extra $s$ bits, we can use a smaller look up table to cover wider input domains. Additionally, since we have known the remaining positive inputs are fairly small, as shown in Figure 6c, we can shift the input range of the look up table to cover more negative input domain. Finally, by multiplying them together, we achieve the exponential results.

## 5 EXPERIMENT

### 5.1 EXPERIMENT SETUP

**Setup.** We choose LLMs including GPT-2 (Radford et al., 2019), Llama2-7B (Touvron et al., 2023), Mistral 8x7B (Jiang et al., 2023), all of which are decoder-only transformer-based language models. We include Mistral 8x7B to demonstrate SOAL framework can work on LLMs with different structures, i.e. Mixture of Expert. Note that, although BERT is commonly used in previous works, it is an encoder-decoder transformer that does not present the phenomenon mentioned above.

| Tokens | Time (s) | | Comm(GB) | | Key Size (GB) | |
|---|---|---|---|---|---|---|
| | Sigma | SOAL | Sigma | SOAL | Sigma | SOAL |
| 64 | 1.097 | **0.852** | 0.370 | **0.36** | 6.806 | **5.938** |
| 128 | 1.936 | **1.493** | 0.824 | **0.783** | 14.292 | **11.915** |
| 256 | 4.166 | **2.996** | 1.983 | **1.819** | 33.191 | **25.873** |
| 512 | 10.048 | **7.577** | 5.302 | **4.648** | 86.686 | **61.830** |
| 1024 | 25.136 | **18.486** | 15.949 | **13.341** | 256.449 | **165.093** |

Table 2: Comparison of time, communication cost and size of generated key for Sigma and SOAL on GPT-2.

| | GPT2 | | Llama2-7B | | Mixtral 8x7B | |
|---|---|---|---|---|---|---|
| | Origin | SOAL | Origin | SOAL | Origin | SOAL |
| WikiText-2 | **36.25** | 37.50 | **5.55** | 5.58 | **6.412** | 6.547 |
| C4 | **33.75** | 34.75 | **6.126** | 6.233 | **3.841** | 3.930 |

Table 3: Perplexity on C4 and WikiText2 datasets across different models.

| | Origin | SOAL |
|---|---|---|
| Arc Challenge ↑ | 0.4334 | **0.4343** |
| Arc Easy ↑ | **0.7635** | 0.7618 |
| HellaSwag ↑ | 0.5713 | **0.5730** |
| PIQA ↑ | **0.7807** | 0.7769 |
| Winograde ↑ | 0.6938 | **0.6993** |
| PPL(WikiText) ↓ | **5.55** | 5.58 |

Table 4: Evaluation of Llama2-7B performace using SOAL framework.

As the goal of SOAL is to show the effectiveness of integrating these ubiquitous phenomena of LLM inference with MPC protocols, we employ ASS and FSS as foundational schemes. For ASS scheme, we mainly use CrypTen (Knott et al., 2021) and adopt truncation protocols from (Santos et al., 2024)[1]. We also include MPCFormer (Cheng et al., 2023) and SecFormer (Luo et al., 2024) with their codes yet changing the truncation method as well. For FSS scheme, we use the Sigma (Gupta et al., 2024) protocol. Our experiments are conducted on two Nvidia A100 GPU with 10GB/s bandwidth. In the experiment, we use $[869, 29901, 22550, 5167, 2073, 29928, 13, 1]$ for Llama2-7B, $[17405, 628, 198, 50256]$ for GPT-2, and $[28705, 13, 262, 1]$ for Mixtral 8x7B.

## 5.2 EFFICIENCY OPTIMIZATION

To evaluate the efficiency of SOAL, we record the time and communication cost to inference one token given a 512-token prompt in Table 1 on different LLMs. As we can see, in the original CrypTen, the cost of non-linear protocols dominates the total cost. With the help of SOAL, the improvement in the non-linear operations is significant. We gain about $2\times$ speedup in SiLU/GeLU operations. We achieve more than $3\times$ faster in SoftMax operation as SOAL can find conformant maxima within a constant round. For normalization, although CrypTen only sets 3 Newton-Raphson iterations for its reciprocal square root protocol, and 8 iterations of Taylor Expansion for the exponential used for the initial guess, the corresponding input domain only covers $(0.1, 145.6)$ with $< 0.1\%$ error, whereas the actual input domain always exceeds such a narrow range. Hence, we follow CrypTen's suggestion, modify the protocol's default configuration to get precise result where we use $0.02$ as the initial guess and use 11 Newton-Raphson iterations. On the other hand, our SOAL framework eliminates the outliers and uses the provided scaling secret $[\![\varphi]\!]_\sigma$ to control the input domain. Therefore, we only need two iterations. In total, SOAL shows nearly $2\times$ efficiency than the original CrypTen protocol across different LLMs.

As the non-linear overhead increases quadratically when the input length gets longer, we conduct the experiment to show the input length's impact on LLM inference in Figure 7. As one can see, due to the benefit of softmax, the longer the input sequence is, the more time can be saved by SOAL. Meanwhile, our method achieves similar speedup as SecFormer (Luo et al., 2024), and becomes faster than MPCFormer (Cheng et al., 2023) when making inference with longer prompt. It is worth noting that we do not require any fine-tuning effort.

SOAL also works on other cryptographic primitives. Table 2 shows SOAL's inference result with Sigma. Although SOAL saves minor cost on the communication, it largely reduces the key size, which need to be transferred for each inference. We as well improve the online inference time.

## 5.3 PERFORMANCE EVALUATION

We use Huggingface's Evaluation library to evaluate SOAL (Llama2-7B) with different benchmarks, including Arc Challenge, Arc Easy (Clark et al., 2018), HellaSwag (Zellers et al., 2019), PIQA (Bisk et al., 2020), Winograde (Sakaguchi et al., 2021). We also use PPL on C4 (Raffel et al., 2020) and

---

[1]The origin implementation of CrypTen uses local truncation, which has a secure flaw pointed out by (Li et al., 2023).

WikiText-2 (Merity et al., 2017) to evaluate other models. Even though MPCFormer and SecFormer fine-tune their models for downstream tasks, their method primarily have 1-2% performance degradation reported in their papers. We don't include their results as they only finetuned the BERT, and GLUE benchmark (Wang et al., 2018) is designed for encoder-decoder LLM.

The performance remains almost the same as the original Llama2-7B, since SOAL only slightly changes the user input instead of changing the entire model weight. Additionally, perplexities over different LLMs are provided in Table 3. The perplexities are close to the origin result, showing that our framework has the robustness to work with LLMs of different structures.

## 6  CONCLUSION

We present SOAL, an efficient MPC framework optimized for decoder-only LLMs. We offer a new direction to improve MPC LLM inference, which involves utilizing LLM's unique properties. We extend the study of outliers, a well-known phenomenon in linear layer, to non-linear layer. SOAL addresses these outlier phenomena and reduces the corresponding input domain. Therefore, we design new non-linear protocols based on it and gain nearly $2\times$ speedup given long inputs on different LLMs. We believe there exist more unique features in LLM, waiting to be explored and exploited.

### ACKNOWLEDGMENTS

This work is supported by Shanghai Qi Zhi Institute Innovation Program. The work of Zhixuan Fang is also supported by Tsinghua University Dushi Program.

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
