# OpenReview forum: "Secure Outlier-Aware Large Language Model Inference"
_ICLR.cc/2026/Conference — ICLR 2026 Poster_

### Official Review · Reviewer_Hgte · 2025-10-21

**Soundness:** 3
**Presentation:** 3
**Contribution:** 3
**Rating:** 6
**Confidence:** 4

**Summary:**

This work introduces a secure inference system that smooths out outliers in order to reduce complexity (bit length) in underlying secure computation. They start from empirical investigation of the outlier issues and show the possibility to constrain their values within a small range by prefixing the prompt. They then propose optimized MPC protocols to evaluate the non-linear primitives on small bit-length inputs. They also propose the "conformant maxima" to compute approximate max values in softmax, which is based on the observation that the most maxima positions locate in a limited number of places.

**Strengths:**

+ I appreciate their effort to identify origins of the outlier issues (and also maxima positions) and propose simple and effective way to reduce such issues in inference, consequently leading to more efficient secure computation on smaller bit length.

+ The proposed protocols seem valid.

+ The proposal is evaluated on both ASS and FSS schemes, which exceeds my expectation.

**Weaknesses:**

- Some claims have no reference support (or might even be invalid, see my questions below).

- The usage of CrypTen is understandable, but it provides misleading results, which should be clearly identified and explained to the readers.

- The experimental comparison is not complete.

**Questions:**

- Give reference for your claim: "Inferencing a 64-token input once using Llama2-7B (Gupta et al., 2022) via CrypTen (Knott et al., 2021) would take 169.76 seconds."

- Your claim "Such inefficiency stems from MPC’s difficulty in handling nonlinear layers" is not completely correct. It is only meaningful because you consider everything implemented in CrypTen, which is not a fair library to use for two-party computation (SOAL focuses on standard two-party computation as you claim). CrypTen is NOT a pure 2PC library because it doesn't include faithful ways of generating 2PC multiplication triples. It should be classified as a "2+1" library where triples are generated by a trusted third party. In pure 2PC secure inference system, the linear layers (mainly matrix multiplication) actually occupies at least half of the overhead (check BOLT[1] or BumbleBee [2] for reference).

- In Figure 1 (c), how can I read from the graph that special tokens are related to outliers?

- For ASS instantiation, how is your 8-bit LUT implemented?

- As I mention above, please clarify in the text that the evaluation results with CrypTen is not pure 2PC. For example, in Table 1, if we evaluate with a "real" 2PC library, the linear layers should contribute much more in the total overhead.

- Should include some recent schemes into the comparison, such as BOLT [1] and BumbleBee [2]. Though these schemes are not implemented with CrypTen, it doesn't affect a direct comparison by taking their results from the papers. For example, communication amount is a metric that normally doesn't change in different libraries, and would give readers a more clear picture on the advantage of your solution.


[1] https://eprint.iacr.org/2023/1893
[2] https://eprint.iacr.org/2023/1678

---

> ### Author Response · Authors · 2025-11-20
> **Response to Reviewer Hgte**
>
> We appreciate the insightful comments from the reviewer. We hope to address your concern with the following response.
>
> **Response to Q1: Give reference for your claim: "Inferencing a 64-token input once using Llama2-7B (Gupta et al., 2022) via CrypTen (Knott et al., 2021) would take 169.76 seconds.**
>
> We thank the reviewer for the careful examination of our claims. We appreciate the opportunity to clarify the source of this data.
>
> We clarify that the figure of 169.76 seconds is an empirical measurement conducted by us specifically for this paper, rather than a citation from existing literature. Since the offical CrypTen did not support Llama model, we adapted the framework to support Llama2-7B architecture to run the actual inference.
>
> Additionally, MPCFormer reports approximately 59 seconds for private inference using BERT-base model with 512 tokens. Considering that BERT-base model has only 0.11B parameters, whereas Llama2-7B has 7B parameters, we believe the magnitdue of this latency is reasonable.
>
> We will revise the sentence to explicitly state the source of such data.
>
> **Response to Q2: 2PC confusion.**
>
> We appreciate the reviewer's experties regarding protocol classifications. We clarify that our setting is a 2PC with a trusted dealer setting. We apologize for using confusing term "2PC with the preprocessing model" in Section 4. The trusted dealer exists in the preprocessing/offline phase, creating the correlated randomness for online phase. Our claim regarding "inefficiency stemming from non-linear layers" specifically refers to the online phase in the 2PC + 1Dealer setting.
>
> We agree that the point reviewer made that in pure 2PC protocols, linear layers can dominate the overhead due to OT extentions or HE. We find that in BumbleBee (Table IV), for GPT-2 with 128 tokens, non-linear layers occupy 36.19\% of the runtime. However, in Table VI, for a ViT-Base model (comparable size) with 197 tokens, this ratio rises to 49.11\%. The proportion of non-linear layers also grows with sequence length in 2PC setting.
>
> We will revise the statements to qualify our claim: "In the online phase of preprocessing-based MPC frameworks, the inefficiency stems from ... " to ensure our motivation is presented precisely.
>
> **Response to Q3: Outliers Relationship in Figure 1 (c).**
>
> We apologize if the visual representation was not immediately intuitive. In the experiment shown in Figure 1(c), we explicitly prefixed the sentence with '.' (period), '$\backslash $n' (newline), and '$\langle s \rangle$' (begin of sentence) tokens. X-axis represent the tokens, Y-axis is the feature dimensions and Z-axis shows the layernorm activation magnitude to the median. There are several peaks appearing at the first three token positions, corresonding to the three special tokens. Comparing to most activations, those peaks are the outliers which is many orders of magnitude larger. On the other hand, values for remaining tokens present as almost flat lines.
>
> **Response to Q4: ASS 8-bit LUT implementation.**
>
> We thank the reviewer for inquiring about the low-level implementation. Our LUT protocol follows the basic LUT protocol from Curl [1], adapted for the CrypTen framework. The core idea is to use correlated randomness generated in the offline phase (by the Trusted Dealer) to mask the input index.
>
> 1. Offline Phase: The dealer generates a random index $r$ and its secret-shared one-hot encoding $[[\mathbf{e}_r ]]$.
> 2. Online Phase: We compute the shift $s = x - r$ and reveal it. Since $r$ is uniform random, $s$ reveals no information about $x$.
> 3. Retrieval: We then cyclically rotate (shift) the secret-shared one-hot vector $[[\mathbf{e}_r]]$ by $s$ positions. By linearity, this results in the one-hot encoding of $x$ (i.e., $[[\mathbf{e}_x ]]$). Finally, we compute the dot product with the table $T$.
>
> [1] Santos, M. B. et al. Curl: Private LLMs through Wavelet-Encoded Look-Up Tables. Preprint at https://eprint.iacr.org/2024/1127 (2024).
>
> **Response to Q5: Clarify that the evaluation results with CrypTen is not pure 2PC.**
>
> Again, we apologize for the usage of confusing term. We acknowledge that both CrypTen and Sigma operate under the 2PC+1Dealer model, where correlated randomness is generated offline by the trusted third party dealer. In such case, the overhead of multiplication is avoid in the online phase.  We will modified the misleading statement '2PC in the preprocessing model' to '2PC+1Dealer model' in Section 4 and Appendix.

---

> ### Author Response · Authors · 2025-11-20
> **Response to Reviewer Hgte (Part 2)**
>
> **Response to Q6: Include BOLT and BumbleBee into the comparison.**
>
> We agree that BOLT and BumbleBee represent the SOTA in pure 2PC inference and are important reference points. We appreciate to position our work relative to them.
>
> While we recognize the value of comparison, directly comparison between secret sharing and HE methods is not proper. HE schemes optimize for lower communicaiton at the cost of higher computation, while secret sharing schemes accept higher communication cost to smaller computation overhead. We offer comparison between SOAL and other works like MPCFormer and SecFormer in Figure 7.
>
> As SOAL's contribution is orthogonal to the underlying cryptographic frameworks, we are willing to explore an implementation based on HE method. We hope to share valid experimental results regarding SOTA HE methods if they become available within the rebuttal window.

---

> > ### Comment · Reviewer_Hgte · 2025-11-27
> >
> > Thank you for the author's response. I am glad to see their clarification about the method, and would like to maintain my current positive rating.

---

> > > ### Author Response · Authors · 2025-12-03
> > >
> > > We apologize that the implementation of SOAL on the BumbleBee framework is not yet complete. However, during the integration of HE methods, we discovered that our ASS SOAL could be further optimized. We plan to complete the HE design and experiments in the near future. Thank you.

---

### Official Review · Reviewer_8jL4 · 2025-10-21

**Soundness:** 2
**Presentation:** 2
**Contribution:** 2
**Rating:** 4
**Confidence:** 4

**Summary:**

SOAL focuses on improving the efficiency of LLM inference under MPC settings. In the past, the main reason for slow inference was the computational complexity of the nonlinear layer. SOAL leverages the statistical properties of activations, maintains the original model weights, and uses a designed trigger token to control the location of outliers, further narrowing the activation range.

**Strengths:**

In the preparation stage, the distribution of nonlinear layer activations is pre-counted, outliers of special tokens are identified, and outliers are locked in predictable positions by prefixing; based on the previously shrunk input, the MPC protocol of RMSNorm, SiLU, and Softmax is reconstructed to achieve acceleration, and the advantages in time and communication volume are reported on gpt2, llama2-7b, and mixtral.

**Weaknesses:**

**1. Idealistic Assumptions:**

The evaluation is conducted in a semi-honest setting, lacks discussion of malicious actors, and provides no security proof.

**2. Lack of Robustness Assessment:**

This method relies on a stable association between activation distributions and specific tokens, but the paper does not verify whether this relationship holds in **multilingual, long-context, or fine-tuned models** scenarios. It also lacks robustness tests against outlier distribution drift and prefix failure.

**Questions:**

1. Is the post-prefix outlier anchor position stable in complex scenarios such as multilingual, long-context, and retrieval-enhanced or multimodal? Has performance been tested under different sampling parameters ($temperature, top-p$)?

2. After incremental training/instruction fine-tuning, is it necessary to re-count special tokens? How does the cost of the Preparation phase versus retraining affect the quantitative benefits of model changes?

---

> ### Author Response · Authors · 2025-11-20
> **Response to Reviewer 8jL4**
>
> We are grateful for the reviewer’s rigorous assessment. We provide clarifications below and hope they alleviate your concerns:
>
> **Response to Q1: Idealistic Assumptions.**
>
> We acknowledge that our evaluation focuses on the semi-honest setting. We would like to clarify the rationale behind the choice and address the concerns regarding security proofs.
>
> First of all, the semi-honest assumption is the standard and prevailing setting for efficienct PPML. Recent works, like CrypTen, Bolt, Iron, Sigma, all operate under this threat model. Malicious security bring a higher-level privacy guarantee to models and inputs, while at the same time introduces higher latency and makes PPML harder to use.
>
> Secondly, SOAL aims to control outliers in LLM and reduce input domain for MPC protocols, and it does not invent new cryptographic primitives. Instead, we build the protocols upon the existed primitives from CrypTen and Sigma. The security of our framework is inherited from the underlying primitives.
>
> **Response to Q2: Lack of Robustness Assessment**
>
> We thank the reviewer for urging a rigorous assessment of our method's robustness in complex scenarios. We fully agree the stability is crital for our protocol optimization. To address this, we have conducted extensive new validation across multilingual, long-context, and multimodal settings, and updated in the Appendix A.
>
> 1. Robustness to Domain Shift
> - Code: We validated the approach on the Llama2-7B model using the `nick007x/github-code-2025' dataset. As shown in Figure 14, the experimental results confirmed that our observations regarding both the special token outliers and conformant maxima remain valid in the code domain.
>   - Multilingual: Multilingual brings low-frequency tokens to prompts, while long-context data uses different rotarty embedding methods during inference. As Llama2-7B only supports English and limits to 4096 tokens, we used Llama3.1-8B instead. For multilingual, we used 8 languanges ['en', 'de', 'fr', 'it', 'pt', 'hi', 'es', 'th'] from huggingface dataset `intfloat/multilingual\_cc\_news'. We show the result in Figure 15.
>   - Long-context: For long-context, we used `jdoo2/Qwen2.5-32B-Instruct\_long\_context \_range80-100 \_train\_data\_100k' dataset and selected inputs longer than 8192 tokens. The special token is $\langle |$begin\_of\_text$|\rangle$ whose token id is 128000 in both cases. The result is shown in Figure 16.
> 2. Robustness to Cross-Modal
> - To further prove the generality of the phenomenon, we extended the validation to a multimodal setting. We conducted experiments using the Qwen2-VL-8B model with a Image-Text-to-Text dataset, `Open-Bee/Honey-Data-15M', and present the result in Figure 17. After filtering out the direct influence of the image tokens, the presense of special token outliers was consistently observed, demonstrating that this characteristic is shared even in multimodal domain.
> 3. Robustness to Finetuned model
> - Llama2-7B-chat is a RLHF finetuned version based on Llama2 7B. We conducted experiments on the instructed version of Llama2 with Redpajama datasets. As shown in Figure 18, the special token outliers persists in the finetuned version.
> 4. Robustness to Random Input
> - We had performed a special test in our Appendix A.1 and Figure 12 to further prove the robustness. In the experiment, we randomly shuffled the input tokens of the input sequences and run the inferences. We found that even with randomized inputs, the outlier observations still held true.
>
> Based on these experiments, we believe the phenomenon is an intrinsic decoder-only LLM features, ensuring robust performance regardless of input domain. We will include these new results in Appendix in the revision.
>
> **Response to Q3: After incremental training/instruction fine-tuning, is it necessary to re-count special tokens? How does the cost of the Preparation phase versus retraining affect the quantitative benefits of model changes.**
>
> During the above experiments, we found that Llama2 7B and Llama2 7B chat share the same specials tokens, so did the Llama3 8B and Llama3.1 8B.  However, as the cost of discovering the special tokens is low, we recommand to re-identify the special tokens after fine-tuning to ensure optimal precision.
>
> SOAL's preparation phase requires only 128 inferences to identify outliers. In the contrast, re-training methods, like MPCFormer using knowledge-distillation, requires not only more GPU memory, but also iterating through the entire dataset for mulitple epochs involving heavy backward propagation. The cost of our preparation phase is orders of magnitude lower than retraining. It serves as a lightweight "calibration" step rather than a training burden, preserving the efficiency benefits of our approach.

---

> > ### Comment · Reviewer_8jL4 · 2025-11-21
> > **follw-up respose**
> >
> > Checked the supplementary experimental results and will increase my score.

---

### Official Review · Reviewer_DABD · 2025-10-24

**Soundness:** 3
**Presentation:** 4
**Contribution:** 3
**Rating:** 6
**Confidence:** 4

**Summary:**

This paper focuses on the problem of secure LLM inference using MPC technologies, specifically 2PC setting. The authors studied the phenomenon of outlier activations during LLM inference and proposed tailored methods to handle outliers for different non-linear functions.  In this way, the input values to non-linear functions, which are typically the bottleneck of MPC-based approaches, can be limited to a narrow range. The authors then used simpler approximations or LUT to improve the efficiency of these non-linear functions. Particullarly, the introduction of conformant maxima avoid the heavy maximum finding. The authors also conducted some experiments against prior works on GPT-2 and LLama-7B models to showcase the effectiveness.

**Strengths:**

- The paper is well-written and easy to follow.
- Using MPC to secure LLM inference is an interesting and important direction.
- The idea of handling activation by prefixing special tokens and using more efficient non-linear functions given a narrow input range is innovative. The introduction of conformant maxima is interesting.
- Extensive experiments have been conducted.

**Weaknesses:**

- The authors focus on 2PC ASS and FSS, lacking the discussion of SOTA HE-based solutions.
- The ASS baseline is mainly Crypten, and lacks comprehensive comparison against SOTA works. For example, MPCFormer, SecFormer and PUMA [1]
- The MPC-side contribution is somewhat limited. The idea is similar to SecFormer and more recent work Nimbus [2], which also proposes distribution-aware non-linear function evaluaiton.
- The description for special-token outlier collection is ambigous. Could the author provide the 'special-toekns' for GPT-2 and Mistral 8x7B as well? According to Figure 1(c), the token '.' (which is believe is not a commonly-defined special token) is regarded as outlier token. What if multile '.' occurs in the prompt? Besides, if the prompt is a long sentence, does that mean all the '.' should be prefixed? In this case, what is the impact to the utility?
- How are the PPL figures in Table 3 calculated? Do the SOAL and original use the same prefixed prompt as inputs?


[1]: PUMA: Secure inference of LLaMA-7B in five minutes. https://arxiv.org/abs/2307.12533

[2]: Nimbus: Secure and Efficient Two-Party Inference for Transformers https://arxiv.org/pdf/2411.15707

**Questions:**

- Could the author clarify whether the setting is exactly 2PC? I believe MPCFormer and Crypten using 2+1 setting, where a TTP is required to handle the generation of some correleated randomness.
- Please refer to the weakness for more questions.

---

> ### Author Response · Authors · 2025-11-20
> **Response to Reviewer DABD**
>
> We thank the reviewer for the constructive comment.
>
>
> **Response to Q1: The authors focus on 2PC ASS and FSS, lacking the discussion of SOTA HE-based solutions.**
>
> We thank the reviewer for raising the discussion point concerning the HE-based solutions. We recognize that ASS, FSS, and HE represent different cryptographic primitives and offer distinct security and efficiency trade-offs.
>
> The goal of SOAL is to offer a new dimension to optimize MPC LLM inference, besides the lines of research dedicating in modifying LLM structure and improving cryptographic primitives. Therefore, we primarily demonstrated the relative performance improvement to its original cryptographic framework. We designed two groups of protocols for ASS and FSS to prove SOAL is orthogonal to the choice of the underlying cryptographic framework..
>
> For HE-based methods, current solutions often require polynomial approximation like CryptoNet, or domain conversion like Iron, Cheetah, Bolt, Nimbus (e.g., between HE and secret sharing), for non-linear operations. SOAL reduces the input range by utilizing LLMs' unique features, therefore, also may bring efficiency to HE-based methods.
>
> **Response to Q2: The ASS baseline is mainly Crypten, and lacks comprehensive comparison against SOTA works. For example, MPCFormer, SecFormer and PUMA**
>
> We thank the reviewer for suggesting comparisons against SOTA baselines. We would like to draw attention to Figure 7, where we presented a comparison against MPCFormer and SecFormer. Our results show SOAL achieving faster speeds than MPCFormer and comparable performance to SecFormer. Crucially, unlike both MPCFormer and SecFormer-which require invasive model structure adjustments and task-specific distillation-SOAL provides a cheaper optimization. Our method achieves high efficiency without altering the pre-trained LLM or requiring costly retraining. This generality is our key advantage.
>
> On the other hand, PUMA utilizes Replicated Secret Sharing (RSS), which is similar yet different to ASS. Besides PUMA's implementatio is based on SecretFlow-SPU framework mainly working on CPU. Direct comparison is not suitable.
>
> We will revise the text to more clearly reference Figure 7.
>
> **Q3: The MPC-side contribution is somewhat limited. The idea is similar to SecFormer and more recent work Nimbus, which also proposes distribution-aware non-linear function evaluaiton.**
>
> We appreciate the reviewer for pointing out SecFormer and Nimbus's contribution to MPC LLM inference.
>
> As far as we know, the operation SecFormer utilized the input distribution is its Goldschmidt reciprocal method in LayerNorm and SoftMax protocols. They divided the denominator value with a constant $\eta=2000/5000$ to get a proper initial guess which previously required complexity non-linear calculations. We believe such division is to constrain certain large denominators as Goldschmidt reciprocal cannot converge with inproper initial guesses.
>
> On the other hand, Nimbus accepts the input distribution as-is and optimizes the polynomial approximation to fit the frequent regions. It does not explicitly handle the outliers, which is critial in our case. Nimbus mainly used the BERT model, which is a encoder-decoder transformers, while we focused on the decoder-only transformer-based LLMs. Although both of them contains similar functions, the causal attention mechanism brings much difference. Nimbus showed the input distribution using the BERT base’s nonlinear functions at the 4th encoder in its Figure 4 ![Imgur](https://imgur.com/SHHC5V0). We also show the input distribution using GPT2 and Llama2-7B's nonlinear functions at different layers in updated Appendix H Figures 18-21. As one can see, the apperance of outliers may hinder us from simply adjusting polynomial approximations to lower-degree in decoder-only transformers.
>
> Both SecFormer and Nimbus passively fit the input distribution, while SOAL regulates the outliers and reduces the range of the remaining inputs. Our method simplifies the problem itself before solving it.

---

> ### Author Response · Authors · 2025-11-20
> **Response to Reviewer DABD (Part 2)**
>
> **Response to Q4: The description for special-token outlier and concern about multiple special tokens as well as impact.**
>
> We thanks the reviewer for the careful observation regarding Figure 1(c) and the request for clarification on outlier tokens. As mentioned in Section 5.1, the identified outlier tokens are [17405, 628, 198, 50256] for GPT-2 and [28705, 13, 262, 1] for Mixtral 8x7B.
>
> For Llama2-7B, the token '.' is the special token. In Figure 1(c), some values at the first token '.' position are much larger than the remaining values, which fits our definition in Section 4.1.1. To address the concern about multiple periods in one sentence, we would like to point out that there are three periods in the input prompt and the outliers appear only at the first period. Therefore, users do not have to prefix all the periods.
>
> **Response to Q5: PPL and experiment input prompts**
>
> For both SOAL and Origin, we break the input sequence into disjoint chunks and add up the decomposed log-likelihoods of each segment independtly, as we want to demonstrate the prefixed tokens do not have large impact on the model's performance. Origin results did not use the prefixed tokens.
>
> **Response to Q6: 2PC+1 setting.**
>
> We appreciate the reviewer raise the ambiguity about MPC setting. We confirm that our setting is indeed a 2PC + 1 Dealer. As described in Section 4, SOAL focused on the standard two-party computation (2PC) in the preprocessing model, where a Trusted Third Party (TTP) is employed during the offline phase to generate correlated randomness for ASS and FSS.
>
> In the revision, we will explicitly adopt the term "2PC with a trusted dealer" to avoid further confusion.

---

> > ### Comment · Reviewer_DABD · 2025-11-27
> >
> > Thank the authors for the detailed response. Most of my concerns have been addressed. However, I have some follow-up questions regarding the special tokens.
> >
> > - What if the prompt contains no special tokens (e.g., “.” or “\n”)? Would the outliers then appear on other tokens (e.g., “the”)?
> > - Does this imply that we could intentionally prepend these identified special tokens to the system prompt in order to avoid activation outliers (since the outlier only occurs where these special tokens first emerge)?

---

> > > ### Author Response · Authors · 2025-12-03
> > >
> > > Thank you for the follow-up questions.
> > >
> > > **Response to no special tokens**
> > >
> > > We conducted a new experiment using randomly selected tokens, ensuring that no special tokens were included. We also suppressed the generation of the *bos* token during tokenization. Under these conditions, outliers were confined solely to the first token position, while values at all subsequent positions remained consistently small and fell within a narrow range.
> > >
> > > **Response to system prompt assumption**
> > >
> > > We appreciate this insightful comment. Indeed, special tokens can be seamlessly integrated into the system prompt to ensure the prepending process feels natural. In our experiments, however, we utilized special tokens alone as prefixed tokens to isolate their effect, ensuring that inference latency measurements were not affected by the overhead of additional tokens. Given the growing criticality of prompt and context engineering, our method is designed to be fully compatible with these paradigms.

---

### Official Review · Reviewer_RtFw · 2025-11-01

**Soundness:** 2
**Presentation:** 2
**Contribution:** 2
**Rating:** 6
**Confidence:** 3

**Summary:**

The paper proposes SOAL (Secure Outlier-Aware Large Language Model Inference), a framework that accelerates privacy-preserving LLM inference under MPC (multi-party computation) by exploiting outlier distributions in non-linear layers. The authors observe that activations in normalization, activation (SiLU), and Softmax layers show heavy-tailed distributions where only a few “outlier” activations dominate. They propose to “control” these outliers by prefixing special tokens in the prompt, thereby reducing the input domain of non-linear MPC protocols. They then design optimized MPC protocols for RMSNorm, SiLU, and Softmax based on the narrowed input range and specific polynomial approximations. Experiments on GPT-2, Llama2-7B, and Mixtral 8×7B show about 2–3× latency reduction in secure inference with nearly no accuracy loss.

**Strengths:**

1. SOAL requires no model retraining or architecture modification, increasing its compatibility with commercial LLM deployments.

2. The paper demonstrates large latency reductions (≈2×–3× on major operations) while maintaining accuracy, which is significant for the secure inference community.

**Weaknesses:**

- The main idea, i.e. using activation range reduction to simplify MPC protocols, draws heavily from prior work in quantization and outlier suppression (e.g., SmoothQuant, Outlier Suppression). The contribution is more engineering-oriented integration than a new theoretical principle.

- The prefix-based outlier control is validated on a fixed set of prompts; the paper does not analyze robustness to domain shift (e.g., multi-language, code, long context). The approach may overfit the profiled activation statistics.

**Questions:**

See weaknesses.

---

> ### Author Response · Authors · 2025-11-20
> **Response to Reviewer RtFw**
>
> We thank the reviewer for the constructive comments and positive assessment. Below, we address the specific concerns.
>
> **Response to Q1: The contribution is more engineering-oriented integration than a new theoretical principle.**
>
> We appreciate the reviewer's insightful comments regarding the novelty of our approach and its relation to existing quantization techniques. We agree that our main idea, leveraging activation range reduction to simplify MPC protocols, is inspired by prior work like SmoothQuant. However, we argue that our contribution goes beyond a mere engineering integration; it introduces a novel paradigm shift in optimizing MPC protocols for LLM Private Inference, while previous works either focus on developing new cryptographic primitives, or modifying ML models to be `MPC-friendly'.
>
> Additionally, prior quantization and outlier suppression works focused primarily on linear layers for clear-text computation and memory efficiency. We extend and validate the special token outliers observation to the inputs of non-linear layers with thorough experiments, which are the dominant bottleneck in MPC protocols. We also designed new non-linear algorithms to fit such range reduction.
>
> What's more, existing empirical observations, such as the Attention Sink phenomenon, alone are insufficient, as shown in Figure 6(c) that about 27\% maxima do no appear at bos token position. We conducted deeper analysis and proposed the Conformant Maxima for achieving substantial range reduction.
>
> **Response to Q2: The prefix-based outlier control is validated on a fixed set of prompts; the paper does not analyze robustness to domain shift (e.g., multi-language, code, long context). The approach may overfit the profiled activation statistics.**
>
> We thank the reviewer for raising the crucial point regarding the robustness of our outlier control to domain shift and the potential for overfitting the profiled activation statistics. We fully agree that robustness across various domains and inputs is essential for a generalized optimization principle.
>
> To address this concern comprehensively, we conducted extensive new experiments covering cross-domain, and cross-modal scenarios. We upload a new version of supplymentary and the experiment results are provided in the corresponding pdf file.
>
> 1. Robustness to Domain Shift
> - Code: We validated the approach on the Llama2-7B model using the `nick007x/github-code-2025' dataset. As shown in Figure 14, the experimental results confirmed that our observations regarding both the special token outliers and conformant maxima remain valid in the code domain.
>   - Multilingual: Multilingual brings low-frequency tokens to prompts, while long-context data uses different rotarty embedding methods during inference. As Llama2-7B only supports English and limits to 4096 tokens, we used Llama3.1-8B instead. For multilingual, we used 8 languanges ['en', 'de', 'fr', 'it', 'pt', 'hi', 'es', 'th'] from huggingface dataset `intfloat/multilingual\_cc\_news'. We show the result in Figure 15.
>   - Long-context: For long-context, we used `jdoo2/Qwen2.5-32B-Instruct\_long\_context \_range80-100 \_train\_data\_100k' dataset and selected inputs longer than 8192 tokens. The special token is $\langle |$begin\_of\_text$|\rangle$ whose token id is 128000 in both cases. The result is shown in Figure 16.
> 2. Robustness to Cross-Modal
> - To further prove the generality of the phenomenon, we extended the validation to a multimodal setting. We conducted experiments using the Qwen2-VL-8B model with a Image-Text-to-Text dataset, `Open-Bee/Honey-Data-15M', and present the result in Figure 17. After filtering out the direct influence of the image tokens, the presense of special token outliers was consistently observed, demonstrating that this characteristic is shared even in multimodal domain.
> 3. Finally, we had performed a special test in our Appendix A.1 and Figure 12, which may comfort the concern about the profiled activation statistics. In that experiment, we randomly shuffled the input tokens of the input sequences. Even with randomized inputs, the outlier observations still held true.
>
> These findings suggest that the outlier behavior is an intrinsic feature developed by LLMs during their training process. This inherent property is what makes our range reduction principle a robust and generalizable optimization. We hope that such results will ease your concern.

---

> > ### Comment · Reviewer_RtFw · 2025-11-27
> > **Comments**
> >
> > Thank you for the clarification. I will keep my original score. As I am not an expert in this area, I would suggest that the AC assign a lower confidence level to my evaluation.

---

### Author Response · Authors · 2025-12-03

Dear Area Chair and Reviewers,

We sincerely thank you for your insightful and constructive feedback. Your comments not only helped us identify areas for clarification but also inspired us to further refine our proposed method. During the implementation of HE method, we were aware that our ASS sigmoid and exponential method could be optimized with small modifications in algorithm.

Therefore, while maintaining our major contribution of utilizing the outlier phenomena  to speed up MPC inference unchanged, we have updated the manuscript with the improvements on detailed protocols and redo the experiments. In detail,

1. We upgraded the sigmoid and exponential methods: We found that the original polynomial approximation of exp could be substituted with local calculation and resharing method, which can save us one more multiplication. We update the original SoftMax's reciprocal method with an efficient MSB and reciprocal protocol.

2. Improved Performance: The updated SoftMax and SiLU method achieves 22.27% and 13.95% speed up improvement on Llama2-7B, compared to the version in the initial submission. Minor improvement is achieved on RMSNorm coming from code refactoring.

Our core idea and contributions remain untouched. FSS-based algorithm is the same. We have added more experiments according to reviewer's suggestion, including different long-context, multilingual, multi-modal scenarios, to prove the robustness. We have modified the confusing term, like '2PC in the preprocessing model'. We believe these updates and improvements may strengthen our paper.

Best regards, Authors

---

### Meta-Review · Area_Chair_wqYt · 2025-12-08

**Summary:**

This paper proposes SOAL (Secure Outlier-Aware Large Language Model Inference), a framework designed to accelerate privacy-preserving LLM inference under Multi-Party Computation (MPC). The core insight is that activation outliers in LLMs, which typically bottleneck non-linear operations in MPC, can be predicted and controlled by prefixing special tokens. By narrowing the input domain for operations like RMSNorm, SiLU, and Softmax, the authors design optimized protocols that significantly reduce latency without requiring model retraining.

The reviewers generally appreciated the practical value of the work, noting that it addresses a critical bottleneck in secure inference without invasive model modifications. Strengths identified include the innovative use of the outlier phenomenon to simplify MPC protocols, the extensive evaluation across different cryptographic schemes (ASS and FSS), and the significant speedups reported (2x-3x). The initial concerns primarily focused on the robustness of the outlier phenomenon across different domains (multilingual, code, long-context), the clarity of the cryptographic setting (2PC vs. 2PC+Dealer), and comparisons with state-of-the-art methods like HE-based solutions. The authors engaged constructively with the reviewers during the rebuttal phase, resolving the major concerns raised by the reviewers.

In summary, due to the practical value of the proposed method, the strong empirical results, and the solid improvements made during the rebuttal, I recommend this paper for acceptance.

**Reviewer Concerns:**

Robustness across domains: Reviewers RtFw and 8jL4 questioned whether the outlier phenomenon holds for multilingual text, code, or long-context inputs. The authors provided comprehensive additional experiments covering code, multilingual, multimodal, and long-context scenarios, demonstrating that the outlier behavior is an intrinsic feature of these models.

Cryptographic Setting Clarity: Reviewers Hgte and DABD pointed out that the paper's description of "2PC" was misleading, as the underlying frameworks (CrypTen/Sigma) rely on a Trusted Third Party (Dealer) for offline randomness generation. The authors acknowledged this and committed to correcting the terminology to "2PC with a trusted dealer."

Special Token Mechanism: Reviewer DABD asked about edge cases where prompts lack special tokens. The authors clarified via new experiments that even with random tokens, outliers persist at the first token position, validating their "outlier-aware" strategy.

Baselines and Comparisons: Reviewers asked for comparisons against SOTA methods like MPCFormer, SecFormer, and HE-based approaches (BOLT, BumbleBee). The authors clarified that SOAL is orthogonal to the underlying crypto primitives and highlighted existing comparisons with MPCFormer/SecFormer where SOAL is competitive or faster without retraining.

**Reviewer Scores:**

Reviewer Hgte, DABD, and RtFw decided to maintain their score. Reviewer 8jL4 explicitly stated to increase the score after reviewing the supplementary robustness experiments.

---

### Decision · Program_Chairs · 2026-01-26

Accept (Poster)